# Exploratory study of staff perceptions of shift safety in the critical care unit and routinely available data on workforce, patient and organisational factors

Clare Leon-Villapalos [1], Mary Wells [2,3] Stephen Brett [1,4]

¹Department of Critical Care, Imperial College Healthcare NHS Trust, London, UK
²Imperial College London, London, UK
³Directorate of Nursing, Imperial College Healthcare NHS Trust, London, UK
⁴Department of Surgery and Cancer, Imperial College London, London, UK

**Correspondence to**
Mrs Clare Leon-Villapalos;
clare.leon-villapalos@nhs.net

## ABSTRACT

**Objectives** To explore bedside professional reported (BPR) perceptions of safety in intensive care staff and the relationships between BPR safety, staffing, patient and work environment characteristics.

**Design** An exploratory study of self-recorded staff perceptions of shift safety and routinely collected data.

**Setting** A large teaching hospital comprising 70 critical care beds.

**Participants** All clinical staff working in adult critical care.

**Interventions** Staff recorded whether their shift felt 'safe, unsafe or very unsafe' for 29 consecutive days. We explored these perceptions and relationships between them and routine data on staffing, patient and environmental characteristics.

**Outcome measures** Relationships between BPR safety and staffing, patient and work environment characteristics.

**Results** 2836 BPR scores were recorded over 29 consecutive days (response rate 57.7%). Perceptions of safety varied between staff, including within the same shift. There was no correlation between perceptions of safety and two measures of staffing: care hours per patient day (r=0.13 p=0.108) and Safecare Allocate (r=−0.19 p=0.013). We found a significant, positive relationship between perceptions of safety and the percentage of level 3 (most severely ill) patients (r=0.32, p=0.0001). There was a significant inverse relationship between perceptions of safety and the percentage of level 1 patients on a shift (r=−0.42, p<0.0001). Perceptions of safety correlated negatively with increased numbers of patients (r=−0.44, p=0.0006) and higher percentage of patients located side rooms (r=0.63, p<0.0001). We found a significant relationship between perceptions of safety and the percentage of staff with a specialist critical care course (r=0.42. p=0.0001).

**Conclusion** Existing staffing models, which are primarily influenced by staff-to-patient ratios, may not be sensitive to patient need. Other factors may be important drivers of staff perceptions of safety and should be explored further.

## INTRODUCTION

Intensive care units (ICUs) are complex and expensive services where staffing is the greatest cost. Inadequate staffing can have serious consequences, including failure to provide fundamental care and patient

### Strengths and limitations of this study

► Our study included staff from all professions working in critical care.
► The study achieved a fair response rate from staff with complete information on patient status and organisational metrics.
► This exploratory study was from different intensive care units, but from the same overarching institution; thus, we cannot be certain how far our observations can be generalised.
► In this study, we were unable to explore the meaning behind why staff felt a particular shift was 'safe' or not.

harm.[1 2] Research suggests that there is an interdependent relationship between the nature and composition of the workforce, work environment and patient outcomes including mortality, in both acute[3–9] and ICU[10–12] settings. The mechanisms that mediate these relationships are incompletely understood.[13] High-quality, patient-centred care requires appropriately skilled and available personnel operating within staffing models that optimise their performance, but this may be difficult to achieve in the context of fiscal and labour market challenges.[13 14]

ICU-specific staffing guidance, such as the UK Guidelines for the Provision of Intensive Care Services,[15] is primarily based on staff-to-patient ratios, and the 'severity' of illness, as defined by the number of organs in failure[16] (online supplementary table 1). The nature and intensity of critical care is, however, changing in response to multiple drivers, including technological advances, more numerous and more complex patients, an increased focus on survivorship and rehabilitation[17] and changing expectations of patients and families; therefore, more complex models for staff deployment may be required.

Current approaches to safe staffing may not be sensitive to actual patient need or to rapid or unpredictable fluctuations in workload. Furthermore, it is unclear how tools to guide staffing establishments translate into staff deployment and staff perceptions of care. ICU workforce guidance[15] also tends towards a uni-professional 'siloed' approach that does not reflect the current multi-professional nature of intensive care[18] or the significance of inter-professional dynamics for productivity.[19][20]

The degree to which existing models of staffing in ICU reflect issues of relevance to staff is also unclear, specifically, staff perceptions of 'safety'—a term used colloquially to reflect an environment, human and physical, which minimises the risk of avoidable adverse events and deficits in care. Recent studies have indicated that perceptions of safety vary within institutions,[21] between individuals[22] and between staff and directors of institutions.[21] Perceived safety may also contribute to staff stress, particularly when there is a discrepancy between expectations of resource and actual resources, including staffing.[23][24] Staff stress is recognised as prevalent and concerning[25] in the ICU workforce and may contribute to reduced quality of care, burnout and poor staff retention.[26]

Relationships between staff perceptions of 'safety' and shift-by-shift variations in staffing metrics, staff deployment, patient and unit factors have not been explored in ICU. There have been few studies which examine the contribution and perspectives of all healthcare professionals within ICU, although this has been identified as a priority.[11][27]

This study, therefore, sought to contribute to our understanding of the complexity of staffing in ICU by:

> Recording staff grading of perceived safety, on a shift-by-shift basis, using a bedside professional reported (BPR) safety estimate (for all professions).
> Exploring the relationships between BPR safety and routinely available data on staffing, patient and work environment factors that influence deployment.

## METHODS

This study was performed at Imperial College Healthcare NHS Trust, across 70 general critical care beds. The three units vary in specialty, size and number of side rooms (online supplementary table 2) but are the part of one institution with shared processes, staffing and staff deployment models. No patient or staff identifiable data were used, and explicit consent was not required. Data were collected over 29 consecutive days during October and November 2018. During the study period, each site was visited or contacted daily by one of the investigators (CL-V), to collect and verify data.

## Patient and public involvement

There were no funds or time allocated for patient or public involvement.

The target population, staff from all professional groups, was involved in informal discussion of the research question, directly participating in the study and

in 'sense checking' the methods of analysis. Results of the study were disseminated to several staff forums, including those for people in leadership positions who direct staffing deployment.

## Data collection and measures
### Bedside professional rating (BPR) of safety

We asked all ICU staff to rate each shift as 'safe, unsafe or very unsafe' using a coloured sticker (green, amber or red). This question was generated from previous work into staff experiences of safety.[22] During staff briefings and daily visits to sites, participants were encouraged to interpret the term 'safe' as their own perception of safe in its normal conversational sense in ICU.

Staff were asked to choose one sticker to rate each shift and put this onto a card labelled with the relevant date and shift (eg, 4 October Night shift). Participants posted the cards in data collection boxes, allowing them to complete the process anonymously, thus, we were not able to identify participants by profession.

To allow us to look for relationships between perceptions of safety and other variables, including staffing, patient and work environment factors, a BPR shift score was created for each shift using the following method; we counted the responses and allocated a score of 1 for each red response, 2 for each amber response and 3 for each green response. We summed the total score and divided this by the number of responses for that shift. This created a mean shift safety score between 1 (very unsafe) and 3 (safe), which reflected the diversity and weighting of responses and is simple to reproduce. This method was discussed with senior nursing staff to confirm that they felt it reasonably summarised perceptions of safety on a given shift.

### Routinely available data on staffing, patient and work environment
*Staffing and workload data*

We recorded the number of staff (including nurses, doctors and allied health professionals) working clinically in the ICU during the study period. Staffing data were extracted from an electronic rostering system and verified with the nurse in charge. Where professional groups did not use electronic rostering, for example, physiotherapists, the data were collected daily and confirmed weekly with the relevant manager. Staff who worked clinically for part of a 12-hour shift were represented as a proportion of a whole-time equivalent (eg, a physiotherapist working for 6 hours on ICU was counted as 0.5).

We calculated workload intensity scores by summing the total number of organs in failure for each day and dividing this by the number of staff working that day. We looked at this by profession and by all professions combined. We used data from the critical care minimum data set[28] to calculate the number of organs each patient had in failure, this was only available by 24-hour period, and therefore, data were collapsed into a 24-hour period.

We recorded the care hours per patient day (CHPPD) from our electronic roster. CHPPD is a measure that

can be used to assess productivity and facilitates benchmarking.[29] We recorded staffing utilisation data from SafeCare in the commercial rostering system provided by Allocate (AllocateSoftware, Richmond, 2017). This uses the composite acuity/dependency scoring system from the Safer Nursing Care Tool[30] (online supplementary table 3A) to provide nursing utilisation reports to support real-time deployment decisions. This is done by calculating required nursing hours (based on acuity and dependency data entered by senior nursing staff) and dividing this figure by the number of rostered nursing hours to produce a percentage of nursing hours utilisation (eg, required hours 126.4, rostered hours 115.00=109.9% utilisation).

This figure is reported in the following (partially) colour coded categories: <90% green, 90.1%–105.1% %, 105% amber, 105.1%–110% and 110.1% red. There is meaning attached to these five categories, so that green is reported as underutilisation while all figures falling into amber and above are reported as overutilisation. In our analysis, we reflected these categories by allocating a number to each so that green=5 through to red=1. This is illustrated in detail in (online supplementary table 3B).

During the study if we found within-shift changes in staff numbers, patient numbers or patient acuity we recorded the status at approximately 05:00 and 17:00 (day shifts). This was a pragmatic decision that reflected the timings of local processes to manage staffing.

### Patient characteristics data

Data regarding the number of level 1, 2 and 3 patients[15] were recorded by visiting or contacting each site daily. In ICU, patients are broadly categorised by the number of organs in failure, with level 3 patients being the most severely ill (online supplementary table 1).

### Work environment data

We recorded the total number of patients and the number of patients in side rooms per shift and confirmed this daily with a senior nurse.

### Data regarding nurse training

We recorded the number of nursing staff on each shift who had completed a postregistration critical care course (CCC), the accuracy of the data were confirmed with local clinical nurse educators. We did not collect staff characteristic data for other professional groups as this is not routinely available, and the smaller numbers of the group sizes would impact on our ability to perform statistical analysis.

### Analysis

Data were collated into an Excel spreadsheet (Microsoft, Redmond, Washington State, USA). We used descriptive statistics to analyse frequency of responses. Data were tested for normality using the D'Agostino-Pearson Omnibus test and non-parametric tests were used where appropriate. We used Spearman's rank correlation to evaluate strength and direction of relationships; regression

lines and 95% CI were plotted for illustrative purposes. Prism (V.8.01 for Windows, GraphPad Software, La Jolla, California, USA) was used throughout.

Although the shift BPR was primarily analysed as a mean for simplicity, we further explored this using both median and a weighted approach (whereby a weighting of 5 was given to very unsafe, 3 to unsafe and 1 to safe). The purpose of this was to allow us to explore the sensitivity of the BPR tool.

## RESULTS

### Exploring BPR perceptions of safety

A total of 2836 BPR responses were received during 29 days of data collection at three sites, we received responses from 170 out of a total of 174 twelve-hour shifts (98% of shifts). The overall response rate, using a denominator of all ICU staff (ie, all staff working on ICU, including those who were transient or only partially based on ICU) was 57.7%. The combined responses are illustrated in figure 1A. Most responses were green, 'safe' (61%), 18% of responses were amber, 'unsafe' and 21% of responses were red, 'very unsafe'. Total responses and responses by site are illustrated in online supplementary table 4.

We noted that staff perceptions frequently varied within the same shift, for only 13% of shifts did all staff who responded report the same perception of safety for their site.

We calculated a BPR shift safety rating, using the methods described previously, which summarised general shift responses and allowed us to look for relationships between reported shift safety and staffing, patient and work environment characteristics. The range of mean shift BPR scores was 1.00–3.00, median 2.63. The distribution of the BPR shift scores for 170 shifts is illustrated in figure 1B.

The BPR tool summarises the overall feeling of a shift by incorporating all responses but does not differentiate between shifts with low and high response rates. To address this, we looked to see if there was a relationship between response rates and perceptions of safety which might reflect a systematic bias, we found a weak relationship between staff perceiving a shift to be safe and a higher response rate (r=0.156, p=0.0426, figure 1C).

### Perceptions of safety and staffing data

We did not find a significant relationship between perceptions of safety and CHPPD (r=0.13 p=0.1080, figure 2A). We found a weak inverse relationship between perceptions of safety and staffing utilisation (SafeCare Allocate) (r=−0.19, p=0.013, figure 2B).

We found a very weak relationship between perceptions of safety and workload 'intensity' (defined as the number of organs in failure/total number of nurses on shift). Looking only at nursing hours, staff reported feeling safer at higher rather than lower intensity workloads (r=0.15 p=0.047). We did not find a significant relationship between these variables when we included doctors' hours

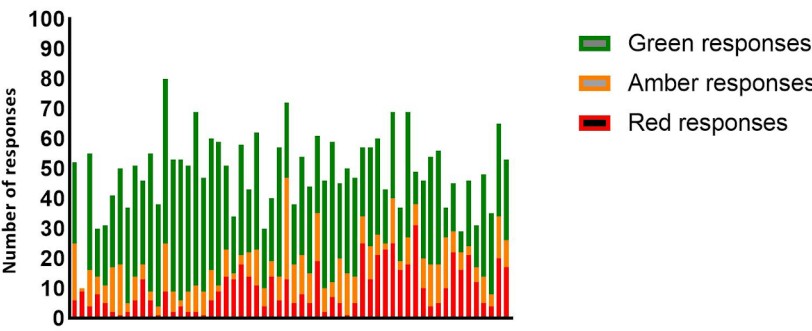

**A** Bedside professional reported shift (BPR)safety October 4th- November 2nd (n=2836).

Green responses
Amber responses
Red responses

Each bar is a 12 hour shift: October 4th day shift- November 1st Night shift

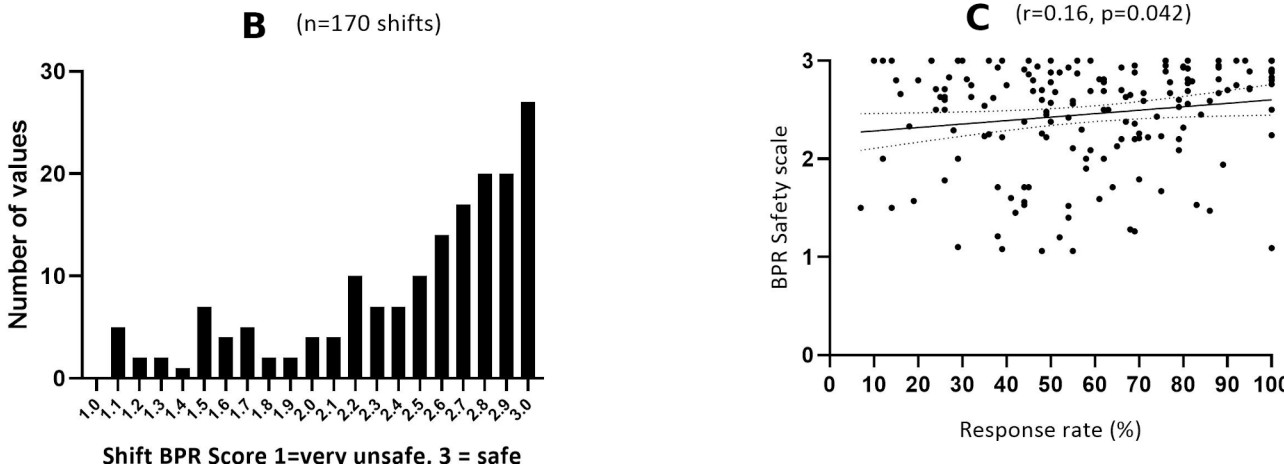

**B** (n=170 shifts)

**C** (r=0.16, p=0.042)

**Figure 1** (A) Bedside professional reported (BPR) safety from 4 October to 2 November (n=2836). (B) Distribution of BPR safety-shift scores (n=170/174 shifts). (C) Relationship between BPR safety and response rates (n=170/174 shifts).

(r=0.14 p=0.73) and allied health professionals (AHP) hours (r=0.13, p=0.09).

### Perceptions of safety and patient characteristics

We found a significant, inverse relationship between perceptions of safety and the percentage of level 1 patients on a shift (p<0.0001, r=−0.42, figure 3A). There was no significant relationship between perceptions of safety and the percentage of level 2 patients (p=0.9729, r=0.003, figure 3B). We found a significant, positive relationship between perceptions of safety and the percentage of level 3 patients (p=0.0001, r=0.32, figure 3C).

### Perceptions of safety and the work environment

We found a significant inverse relationship between perceptions of safety and numbers of patients in the largest unit (unit 1) where the number of patients ranged between 27 and 32 (figure 4A, r=−0.44, p=0.0006). We did not find a significant relationship in the smaller units (where numbers of patients were between 11–16 and 17–23, respectively, figure 4B,C). We also found a

significant inverse relationship between perceptions of shift safety and the percentage of patients who were occupying side rooms in a given shift (r=−0.45, p<0.0001, figure 4D).

We found a significant, positive relationship between the percentage of nursing staff who had a critical care course on each shift and perceptions of safety (r=0.63, p<0.0001, figure 5). We also looked at day versus night scores and there was robustly no difference between perceptions of safety .

These analyses were repeated using both median scores and a score that gave additional weighting to 'unsafe' responses; we found only minor differences in the relationships using these methodologies. These are summarised in online supplementary table 5.

### DISCUSSION

Our study sought to explore an estimate of staff perceptions of shift safety for all professions in ICU on a

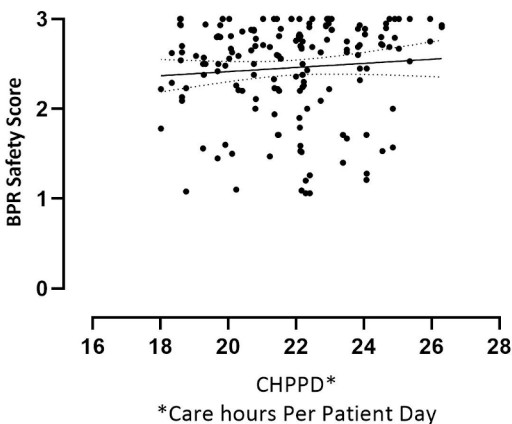

**A** (r= 0.13, p=0.1080)

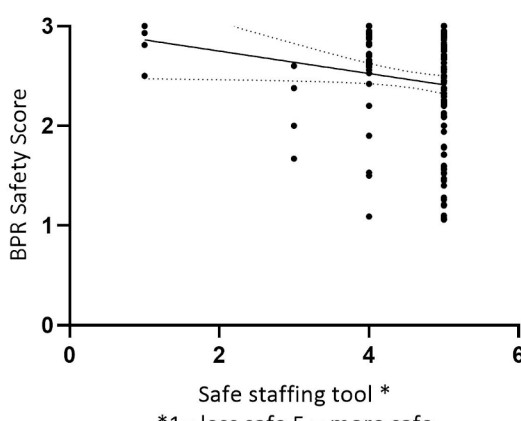

**B** (r=0.19, p=0.013)

**Figure 2** (A) Relationship between bedside professional reported (BPR) shift safety and care hours per patient day (n=170/174 shifts). (B) Relationship between BPR shift safety and safe staffing 'score' 1=staff overutilisation red, 5 = green staff underutilisation (n=170/174 shifts).

shift-by-shift basis; we termed this BPR safety. We also explored the relationships between this estimate and other factors, including routinely available data on staffing, patient and work environment factors that influence deployment. Staff perceptions of safety are important as they may reflect safety culture[31] and be associated with

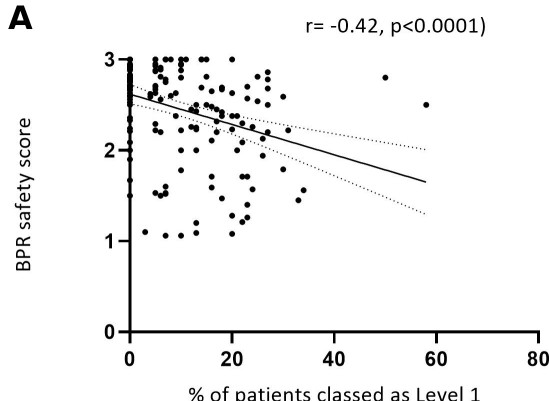

**A** r= -0.42, p<0.0001)

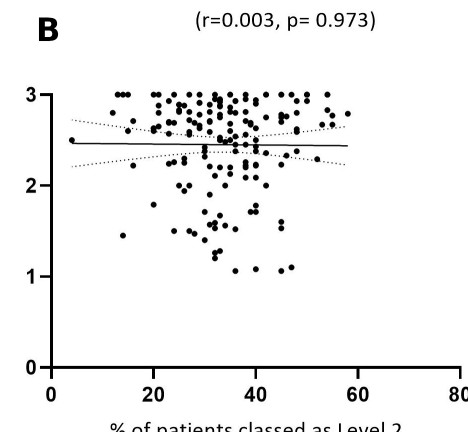

**B** (r=0.003, p= 0.973)

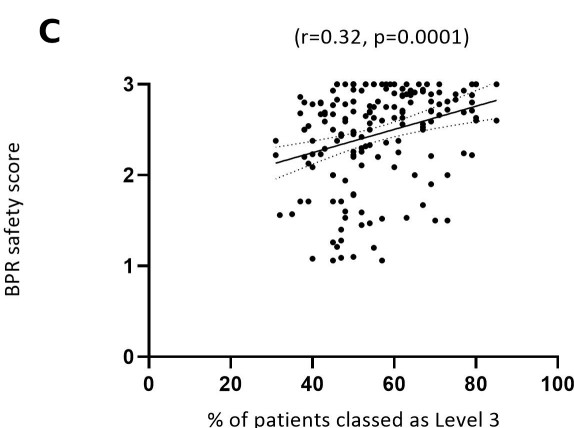

**C** (r=0.32, p=0.0001)

**Figure 3** (A–C) Relationship between bedside professional reported (BPR) shift safety and volumes of acute and less acute patients: level 1, level 2 and level 3 (n=170/174 shifts).

**A**  **Unit 1** (r= -0.44, p=0.0006)

BPR safety scale

Number of patients

**B**  **Unit 2** (r=-0.07, p=0.63)

BPR safety scale

Number of patients

**C**  **Unit 3** (r=-0.13, p= 0.34)

BPR safety scale

Number of patients

**D**  (r= -0.45, p= <0.0001)

BPR Safety Score

% of patients in a side room

**Figure 4** (A–C) Relationships between bedside professional reported (BPR) shift safety and number of patients in unit 1 (n=58), 2 (n=56) and 3 (n=56), (n=58). (D) Relationship between BPR perceptions of shift safety and the percentage of patients in a side room (n=170/174 shifts).

the risk of adverse events,[32] including risks not identified by organisational processes.[1] We observed that staff found the BPR tool easy to implement and seemed acceptable to use (although no formal assessment of acceptability was made); previous pilots using more ambitious methods had proved unsuccessful due to poor engagement. The BPR's simplicity and fair completion rates are strengths and are

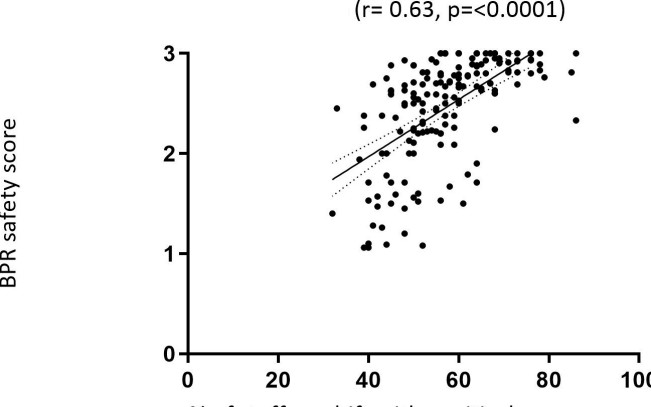

(r= 0.63, p=<0.0001)

BPR safety score

% of staff on shift with a critical care course

**Figure 5** Relationship between bedside professional reported (BPR) shift safety and the percentage of staff on a shift with a critical care course (n=170/174 shifts).

important features of safety tools[31]; it also yields unit level data, which could be used to inform local discussion and action. We are not aware of any studies of similar tools in ICU, and although further validation of the BPR tool is needed, the relationships we found were plausible. The very weak relationship we found between higher response rates and perceived safety indicates that we cannot rule out a systematic bias, with higher response rates on 'safer' shifts. However, neither the strength or direction of the relationship is pronounced. Exploring the data using a median-based or weighted approach to BPR did not alter the observed relationships (online supplementary table 5). Organisations wishing to create a more safety sensitive system might consider further adjustments to weighting.

For most shifts during the study period, we found heterogeneity in reported perceptions of safety. Similarly, different 'risk horizons' between and within staff groups on ICU, depending on team and environmental factors, have been previously described.[22] This may support an argument that training for those with leadership roles in ICU should explicitly address heterogeneity of experience, perceptions of safety and anxiety among ICU staff.

Staff recorded their perceptions of shift safety; being involved in an enterprise which is not felt by the individual

to be 'safe' may be a source of stress. This was reflected in the way some staff responses seemed to be framed (which included annotations and pictures, online supplementary figure 1). We found, during data collection, that staff described the stickers as 'cathartic' suggesting that the BPR facilitated the articulation of feelings that were not captured elsewhere. It is reasonable to suggest that being able to communicate concerns about safety to someone who will listen may be important for staff well-being and trust in leadership. This may be of particular importance in ICU, an environment where staff stress and burnout are areas of concern.[25 33]

Perceptions of shift safety were not well correlated with some nursing staffing metrics (NB deployment metrics for AHPs and medical staff are not currently in use at our organisation.). The lack of relationship between these organisational measures and staff perceptions reinforces the importance of triangulating staff deployment decisions with professional judgement and suggests organisations should be cautious in relying solely on mathematically derived models. This may be one of the reasons that CHPPD is not consistently accepted by nurse directors[34] and it is already noted that SafeCare Allocate has not been validated for use in deployment.[35] Although the use of staffing benchmarks to prevent the erosion of staffing numbers is likely to remain important, the findings of this study suggest that more sophisticated thinking is needed to inform staffing models.

Our study found a significant relationship between patient acuity and perceptions of safety. Perhaps counterintuitively, more staff reported feeling safe when there were larger numbers of severely unwell (level 3) patients, and reported feeling shifts to be 'unsafe' when there were larger numbers of low acuity patients (level 1). We did not specifically explore what mediated this relationship, but possible explanations include: that existing ICU staff deployment models are more sensitive to need for high acuity than for lower acuity patients; that staff are less confident in caring for this patient group; or that they are a marker for other workflow stressors, such as throughput or a very dynamic workload. This finding may indicate that uni-professional approaches that allocate fewer nursing hours to less severely unwell patients (where severity of illness based on categories of organ failure) may not match staff perceptions of patient need.

We found that size of unit and an increased proportion of patients located in side rooms were significantly related to staff reporting the shift as feeling less safe. We did not explore what mediated these relationships; possible explanations might include the impact of larger volumes of patients on individuals' capacity so that there may be a degree of activity which is beyond most staff's comfort zone. In the largest unit, processes were starting to be put in place to organise this as two smaller units but these were not fully embedded during the study period. Side rooms are increasingly common in ICU due to the benefits of improved patient privacy, confidentiality and infection control. However, there is little research into the impact on staff. Recent studies conducted in acute wards suggest that trade-offs for nursing staff include increased staff anxiety due to decreased surveillance and fewer opportunities for informal education, role modelling and interaction.[36 37] Our findings suggest that there may be value in exploring the relationships between staff perceptions of safety and unit design to identify modifiable factors, as larger units and side rooms become increasingly common features of ICUs.

The strong positive relationship between perceptions of safety and the percentage of staff on each shift with a CCC supports guidance that this should be a benchmarked target[15] and suggests that there may be value in training local establishments to a level which enables this to be achieved in daily staffing rotas. Our finding is consistent with the evidence that increased nursing education (to Bachelor level) is associated with a reduced likelihood of patients dying.[4] It is also supported by recent findings,[38] which indicated that strain and burnout in ICU nurses are not inevitable and could be modified by an intensive education programme which included situational role play, simulation and debriefing.

## LIMITATIONS
### Our study had several limitations
While the study included three separate units, it was conducted at a single NHS Trust. Findings therefore may reflect factors unique to our culture and organisation; however, the volume of observations and range of specialities suggest that our data may have broader applicability. We did not perform subgroup analysis which may have yielded differences between units as the smaller groups would have been insufficient for statistical analysis.

This was an exploratory study and the first use of the BPR scale, which although not formally 'validated' seemed to have face validity with participants and senior staff. The intensity-derived data might not be sensitive to variations within a given shift or between sites. Furthermore, although it seems intuitively obvious, we cannot know for certain that working on a shift experienced as 'unsafe' causes anxiety. We did not attempt to capture anxiety because in a preliminary unpublished study, we found that attempting to acquire too much data at the end of a busy shift produced poor survey return and we were keen to avoid undermining the primary objective.

The question 'How safe was your shift' was taken from associated work undertaken previously on our unit. The initial unpublished pilot was perceived as too complex and did not gain traction; we therefore reduced the original 5-point response options to 'safe, unsafe and very unsafe'. During the data collection, we noted that staff often annotated their responses with extra stickers, drawings or comments, indicating that three responses did not always allow them to express fully the depth or range of feelings, which seemed more important at the 'unsafe' end. Equally it could be argued that there is no meaningful difference between unsafe and very unsafe and

that these categories could be collapsed; we therefore felt the option we selected was reasonable.

Early conversations with staff indicated that many staff did not want to indicate their professional group when reporting their perception of shift safety, as they felt this might compromise anonymity; we therefore did not ask for this information. This meant we were not able to analyse responses by the professional group. While this information would have brought valuable insights, it may have impacted on the response rates. Subsequent informal discussions have suggested that, having gained confidence in the tool and motivation behind the research project, staff would be willing to indicate their profession in future studies.

Despite a satisfactory response rate, we did not receive responses from all staff and therefore do not know whether the perceptions of staff who did not engage are different to those who did.

Most significantly, we do not know what lay behind different individual feelings of safety or what might mediate the relationships we identified between perceptions of safety and the characteristics discussed. Further qualitative work is currently being undertaken to address this.

## IMPLICATIONS

Understanding how to deploy the multi-professional team in a way that sensitively matches patient need in ICU is challenging and there is limited evidence available to inform guidance. Incorporating staff perceptions may be useful organisationally and may help identify modifiable factors that impact on staff stress and burnout.

The findings of our exploratory study suggest that

► A BPR tool is simple to use and seems acceptable to staff; it may capture near real-time staff perceptions of safety.
► Perceptions of safety vary between individuals on the same shift and this should be factored into the way teams are trained and led.
► Perceptions of safety may not align to some organisational staffing metrics, reinforcing the need to augment these with professional judgement and other measures.
► The composition of the workforce is important to staff perceptions of safety and our findings support the provision of specialist postregistration critical care course training for nursing staff.
► Further research could usefully focus on staff deployment for patients classified as less severely unwell and the impact of the size and layout of ICUs.

To our knowledge, this is the first study to include allied health professionals in addition to nurses and doctors to explore perceptions of safety and staff deployment in ICU. This approach may yield results that better reflect the reality of how care is delivered in ICU and can also accommodate emerging roles such as nursing associates and advanced critical care practitioners. This study has

relevance nationally; the factors explored reflect, or are similar to, guidance followed by most UK ICUs. The findings may be of interest outside of the UK since, while processes vary, there are shared concerns regarding workforce deployment, supply, costs and burnout.

**Contributors** We have checked the policy on authorship and we are fully satisfied that all authors fulfill the criteria of authorship and that no one else who fulfills the criteria has been excluded as an author. CL-V and SB conceived the idea for the study. All authors contributed to the funding application, design and execution of the study. CL-V, SB and MW co-designed and planned the methodological approach to data collection, use of routine data and statistical analysis. CL-V collected the data and used the statistical package PRISMA to analyse data, with close involvement from SB and MW, who helped make decisions about the appropriate statistical tests and processes to use. All authors contributed to the review, analysis and interpretation of data, and discussed and decided upon the focus and content of the paper and the presentation of results. CL-V wrote an initial draft of the paper which was then revised critically and re-drafted by all authors (CL-V MW and SB). CL-V, SB and MW all made substantial contributions to the design, analysis, interpretation and writing up of this study, and have approved the final manuscript. All authors (CL-V, SB and MW) agree to be accountable for all aspects of the work.

**Funding** This work was supported by Imperial Health Charity, National Institute of Health Research (NIHR) Imperial Biomedical Research Centre funded predoctoral research grant (grant number RF18\100007).

**Competing interests** None declared.

**Patient and public involvement** Patients and/or the public were not involved in the design, or conduct, or reporting or dissemination plans of this research.

**Patient consent for publication** Not required.

**Ethics approval** UK Health Research Authority approval was obtained (ID249248).

**Provenance and peer review** Not commissioned; externally peer reviewed.

**Data availability statement** Data are available in a public, open access repository. Extra data can be accessed via the Dryad data repository at http://datadryad.org/ with the doi: 10.5061/dryad.v9s4mw6s7.

**ORCID iDs**
Clare Leon-Villapalos http://orcid.org/0000-0002-1610-0319
Mary Wells http://orcid.org/0000-0001-5789-2773
Stephen Brett http://orcid.org/0000-0003-4545-8413

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
