## [Reviewer comments · BMJ Open]

ARTICLE DETAILS

TITLE (PROVISIONAL)	An exploratory study of staff perceptions of shift safety in the critical care unit and routinely available data on workforce, patient and organisational factors.
AUTHORS	Leon-Villapalos, Clare; Wells, Mary; Brett, Stephen

VERSION 1 – REVIEW

REVIEWER	Rene Schwendimann University Hospital of Basel Switzerland
REVIEW RETURNED	19-Sep-2019

GENERAL COMMENTS	In their paper, the authors addressed the important issue of shift safety in the critical care setting considering health professional perceptions among other factors. Overall, the paper is well written, and easy to follow regarding methods used, reported results and their reflections on findings with regard to the literature and own interpretations. I have few rather minor comments and suggestions to consider. Page 3, line 20: "Inadequate staffing can have serious consequences (Ref)." Provide 2-3 examples. Lines 22-23: "The mechanisms that mediate these relationships (...)." Did you consider research on quality of the work environment/Nurse work index, e.g. from Aiken et al.? Line 52: "(...) particularly when there is a discrepancy between expectations (...)." Expectations of what? Please specify. Page 4, line 21: "During the study period, each site was visited or (...)." visited/contacted by whom? Please specify. Page 5, line 33: When defining ranges for coding you wrote: "90.1%-104.9%, 105% Amber, 105.1%-109.9%, 110% Red." Here I am confused. Why you do not write "90.1%-105% Amber, 105.1%-110% Red?" Lines 57-60: Here I would expect some information about the other professionals (e.g. physicians), since you claim including all professions (see pg 4, line 6; pg 10, lines 31-32). Page 7, lines 36-37: It may read "We also explored the relationships between (...)."  Line 59: "Both studies support an argument that (...)." In the sentence before, you refer to one other study, which is the second study. Please specify. Page 8, line 14: "important for staff wellbeing." In addition to this argument, you can add "and trust in leadership." Page 9, lines 10-11: "(...) can be modified by an intensive education program." I wonder what education program you have in mind. Especially, what for professionals are enabled by such a program. Please specify.
---

	Line 30: You mention "a preliminary study". Add reference or state "unpublished".
--	---

REVIEWER	Marty van Beuzekom Leiden University Medical Center The Netherlands
REVIEW RETURNED	01-Oct-2019

GENERAL COMMENTS	In general: The study is about exploring, but so many data and correlations are presented that the question Exploring staff perceptions or safety for all professions, on a shift by shift basis, using a bedside professional reported (BPR) safety estimate is not clearly answered, Although the authors indicate this in the Limitations of the study to be unable to explore the meaning behind why staff felt a particular shift was "safe" or not. In a next version of the manuscript It is important. An exploratory study of staff perceptions of shift safety in the critical care unit and routinely available data on workforce, patient and organisational factors. Methods Although the results do not make a distinction between professional groups, for the interpretation of the results I advise to describe where staff consists of. On page 5 something is said about physiotherapists, but that is not described earlier. Results I think Figure 1 is not mentioned in the text I have a question: Was examined whether there is a difference between day and night shift and the perception on safety because of the adverse effects or sleep deprivation and fatigue on staff. Interesting to know if this influences the perception of safety. "We noted that staff perceptions frequently varied within the same shift; for only 13% of shifts did all staff who responded report the same perception of safety for their site" What does this mean for performed statistics and the interpretation of the results? The distribution of the BPR shift scores for 170 shifts is illustrated in Figure 1b. But what is the significance of the figure presented that 1.8 times occurs 8. Describing the results below is confusing, there is talk about units (Table 4) and level patients, but you do not know how many, for example, level 1 patients were on unit 1, which makes it difficult to interpret the results "Most responses were green "safe" (61%), 18 % of responses were amber, "unsafe" and 21% of responses were red, "very unsafe". Total responses and responses by site are illustrated in Supplementary Table 4. I think this is an important table Red (very unsafe) for Unit 1 37%, for unit 2: 5% and for Unit 3: 10%. We found a significant, inverse relationship between perceptions of safety and the percentage of Level 1 patients on a shift ($p < 0.0001$, $r = -0.42$ Figure 3a)
---

	We found a significant inverse relationship between perceptions of safety and numbers of patients in the largest unit (Unit 1) where number of patients ranged between 27 and 32 (figure 4a $r=-0.44$, $p<0.0006$). You can also interpret that from table 4. We did not find a significant relationship between these variables when we included doctors' hours. This is the first time doctors are mentioned, they are not described by characteristic data. In general many correlations are described and explained in figures, but what is the significance of this, moreover, they are usually moderate correlations with the exception of perception or safety and patient characteristics. Supplementary Table 5 is easier to read than all figures.
--	---

VERSION 1 – AUTHOR RESPONSE

Comment	Response	Page and Line in marked copy
Reviewer One		
Page 3, line 20: "Inadequate staffing can have serious consequences (Ref)." Provide 2-3 examples.	We have added in more detail in response to this comment. We also refer to the examples cited in references 3-11.	Page 3, line 11
Lines 22-23: "The mechanisms that mediate these relationships (...)." Did you consider research on quality of the work environment/Nurse work index, e.g. from Aiken et al.?	We have adopted the Reviewers' recommendation and included Aikens' conclusions on work environment. Citations 4 and 5 also refer to Aikens' study of workforce composition which we believe to be the most relevant to this paper.	Page 3, line 13
Line 52: "(...) particularly when there is a discrepancy between expectations (...)." Expectations of what? Please specify.	We thank the Reviewer for identifying this and have included more detail.	Page 3, line 38
Page 4, line 21: "During the study period, each site was visited or (...)."	The amendment has been made.	Page 4, line 17

visited/contacted by whom? Please specify.		
Page 5, line 33: When defining ranges for coding you wrote: “90.1%-104.9%, 105% Amber, 105.1%-109.9%, 110% Red.” Here I am confused. Why you do not write “90.1%-105% Amber, 105.1%-110% Red	We thank the Reviewer for this helpful suggestion and have adopted his recommendations. This does not change the findings but is a clearer way of expressing these.	Page 5, line 32
Lines 57-60: Here I would expect some information about the other professionals (e.g. physicians), since you claim including all professions (see pg 4, line 6; pg 10, lines 31-32).	We accept that the text was confusing. We have added clarity in two ways. Firstly, by detailing in page 5 line 8 that we collected number of staff for all professions. Secondly we have changed the title from Staff Characteristics data to a more specific one “Data regarding nurse training” , line 2 We chose to measure this as it was routinely available; the smaller size of other staffing groups meant this data could not usefully be broken down further.	Page 5, line 8 Page 6 line 3
Page 7, lines 36-37: It may read “We also explored the relationships between (...).”	We thank the Reviewer for this helpful rewording and have made this adjustment.	Page 7 line 39
Line 59: “Both studies support an argument that (...).” In the sentence before, you refer to one other study, which is the second study. Please specify.	We thank the Reviewer for identifying this, we accept our original wording was unclear and have changed this.	Page 8, line 8
Page 8, line 14: “important for staff wellbeing.” In addition to this argument, you can add “and trust in leadership.”	We thank the Reviewer for this suggestion and have included this suggestion.	Page 8, line 17

Page 9, lines 10-11: “(…) can be modified by an intensive education program.” I wonder what education program you have in mind. Especially, what for professionals are enabled by such a program. Please specify.	We have included more detail regarding the education program that we have referenced.	Page 9, lines 13-15
Line 30: You mention “a preliminary study”. Add reference or state “unpublished”.	We have amended this sentence to include the word unpublished.	Page 9 , line 29
Reviewer Two		
General Comment The study is about exploring, but so many data and correlations are presented that the question Exploring staff perceptions or safety for all professions, on a shift by shift basis, using a bedside professional reported (BPR) safety estimate is not clearly answered, Although the authors indicate this in the Limitations of the study to be unable to explore the meaning behind why staff felt a particular shift was “safe” or not. In a next version of the manuscript It is important.	We acknowledge that the term “explore” can be interpreted in several ways. We have taken the opportunity to clarify this by altering the wording of the first aim objective to “record” and changing the syntax to aid clarity We consider this work exploratory in that it is looking to see if there are relationships between staff perceptions of safety and other factors. This study identifies relationships which may be of interest to the wider community.	Page 4, line 5
Methods Although the results do not make a distinction between professional groups, for the interpretation of the results I advise to	We thank the Reviewer for this comment. We have added more detail to the methods section to make it explicit that staff recorded their perceptions anonymously including not stating their professional group.	Page 4, line 38

describe where staff consists of. On page 5 something is said about physiotherapists , but that is not described earlier.	We address in the limitations that staff did not initially want to record their profession on their responses.	
Results I think figure 1 is not mentioned in the text	Figures 1a 1b and 1c are mentioned in the text on page 6 I have highlighted these.	Page 6 Line 34, 42 and 48
I have a question: Was examined whether there is a difference between day and night shift and the perception on safety because of the adverse effects or sleep deprivation and fatigue on staff. Interesting to know if this influences the perception of safety.	We thank Reviewer 2 for this suggestion. We have since analysed and found that there is no difference between scores during the day versus night.	Page 7 line 30-32
“We noted that staff perceptions frequently varied within the same shift; for only 13% of shifts did all staff who responded report the same perception of safety for their site” What does this mean for performed statistics and the interpretation of the results?	We interpreted the Reviewers comment to mean that he was anxious about the statistical resilience of this observation. In fact, we analysed the data in a number of different ways and have presented these analyses in supplementary Table 5. If the Reviewer’s comment referred to the heterogeneity of response within shifts, this is already discussed in page 8 line 10.	Page 6, Line 19-21.
The distribution of the BPR shift scores for 170 shifts is illustrated in Figure 1b. But what is the significance of the figure presented that 1.8 times occurs 8.	We have reviewed figure 1b and do not understand the question posed.	
Describing the results below is confusing, there is talk about units (Table 4) and level patients, but you do not know how many, for example, level 1	We acknowledge the description of the hospital structure could be confusing. The units are wards within a single institution, though, they vary by speciality they share staff, policies and deployment and processes.	Page 4 Line 13

patients were on unit 1, which makes it difficult to interpret the results “Most responses were green “safe” (61%), 18 % of responses were amber, “unsafe” and 21% of responses were red, “very unsafe”. Total responses and responses by site are illustrated in Supplementary Table 4. I think this is an important table Red (very unsafe) for Unit 1 37%, for unit 2: 5% and for Unit 3: 10%. We found a significant, inverse relationship between perceptions of safety and the percentage of Level 1 patients on a shift ($p < 0.0001$, $r = -0.42$ figure 3a) We found a significant inverse relationship between perceptions of safety and numbers of patients in the largest unit (Unit 1) where number of patients ranged between 27 and 32 (figure 4a $r = -0.44$, $p < 0.006$). You can also interpret that from table 4	We have reworded the methods section to clarify this. We were primarily interested in the relationships between staff perceptions of safety and levels of patients . During the study period there were higher numbers of Level of 1 patients in one of the wards , however we were not able to perform linear regression or more complex mathematics to separate out the impact due to the very uneven distribution of Level 1 patients between wards. Thus we cannot comment on the statistical interaction of “ward” and “level of care”. We did not perform subgroup analysis that is perhaps suggested as this would have resulted in small subgroups, insufficient for statistical analysis. We have added an acknowledgement of this to the limitations section. This is something we believe other researchers may want to explore. We feel, as raised in the discussion, that the results indicate that staff deployment by ratios, informed by the numbers of organs in failure, does not seem to hold.	Page 9 Line 21-23
We did not find a significant relationship	The Reviewers comments highlight that we have not signposted clearly enough in the	page 5 Lines 8 and 18

between these variables when we included doctors' hours. This is the first-time doctors are mentioned, they are not described by characteristic data.	methods section where we included other staff: We have added more detail into page 5 lines 8 and 18. We have also reworded the subheading "staff characteristic data" to "Data regarding nurse training" page 6 , line 3 to give greater clarity .	page 6 Line 3
In general many correlations are described and explained in figures, but what is the significance of this, moreover, they are usually moderate correlations with the exception of perception or safety and patient characteristics. Supplementary Table 5 is easier to read than all figures.	We have used figures as we feel most clearly displays the original data and is therefore the most transparent method. We have included Tables in the supplementary data to summarise the findings for the readers convenience. We have also re-ordered and clarified the bullet points in implications to emphasize the significance of the findings more intuitively to the reader.	Page 10 Line 16-25